# A Retrospective Review of 28 Cases of Pediatric Malignant Renal Tumors at a Single Institution

**DOI:** 10.3390/diseases13020040

**Published:** 2025-01-31

**Authors:** Takafumi Kawano, Koshiro Sugita, Ryuta Masuya, Nanako Nishida, Ayaka Nagano, Masakazu Murakami, Keisuke Yano, Shun Onishi, Toshio Harumatsu, Tatsuro Nakamura, Koji Yamada, Waka Yamada, Mitsuru Muto, Yuichi Kodama, Takuro Nishikawa, Tatsuru Kaji, Yasuhiro Okamoto, Satoshi Ieiri

**Affiliations:** 1Department of Pediatric Surgery, Research Field in Medical and Health Sciences, Medical and Dental Area, Research and Education Assembly, Kagoshima University, Kagoshima 890-8520, Japan; kawano-t@m2.kufm.kagoshima-u.ac.jp (T.K.); ksugita@m.kufm.kagoshima-u.ac.jp (K.S.); nanakonishida7@gmail.com (N.N.); ayakanagano05@gmail.com (A.N.); k6126906@m.kufm.kagoshima-u.ac.jp (M.M.); yano-k@m.kufm.kagoshima-u.ac.jp (K.Y.); sonishi@m3.kufm.kagoshima-u.ac.jp (S.O.); harumatsu1040@gmail.com (T.H.); k-yamada@m2.kufm.kagoshima-u.ac.jp (K.Y.); waka1128@m2.kufm.kagoshima-u.ac.jp (W.Y.); mitsuru@m3.kufm.kagoshima-u.ac.jp (M.M.); 2Department of Gastrointestinal Surgery and Surgical Oncology, Ehime University Graduate School of Medicine, Matsuyama 791-0295, Japan; masuya.ryuta.pa@ehime-u.ac.jp; 3Department of Pediatrics, Kagoshima University School of Medical and Dental Science, Kagoshima University, Kagoshima 890-8520, Japan; ta2low1225@yahoo.co.jp (T.N.); yuichik18@gmail.com (Y.K.); adu44150@ams.odn.ne.jp (T.N.); okamoto@m2.kufm.kagoshima-u.ac.jp (Y.O.); 4Department of Pediatric Surgery, Kurume University School of Medicine, Kurume 830-0011, Japan; kaji_tatsuru@kurume-u.ac.jp

**Keywords:** Wilms’ tumor, malignant rhabdoid tumor, renal cell carcinoma, children

## Abstract

Advances in treatment have dramatically improved the outcomes of pediatric renal malignancies. We reviewed cases of renal malignant tumors that were managed in our institution. The patients’ background factors, pathological diagnoses, stages, outcomes and late complications were retrospectively reviewed using medical records of 28 patients with renal tumors who were treated at our institution from 1984 to 2022. Wilms’ tumors were recognized in 24 patients (85.7%), all of whom had favorable histology. Wilms’ tumors were Stage I in six patients (6/24; 25.0%), Stage II in nine patients (9/24; 37.5%), Stage III in five patients (5/24; 20.8%), Stage IV in two patients (2/24; 8.3%), and Stage V in two patients (2/24; 8.3%). Two patients (7.1%) with clear cell sarcoma of the kidney both had Stage I disease. One patient had Stage IV rhabdoid sarcoma of the kidney (3.5%), and one had Stage IV renal cell carcinoma (3.5%). The overall 5-year survival rate was 85.2% for all renal malignancies. Late complications included chronic renal failure in four patients (14.2%). The outcomes are comparable to those reported previously. However, the prognosis of MRTK and renal cell carcinoma remained poor in advanced cases; thus, another therapeutic protocol should be established.

## 1. Introduction

Renal tumors are the most common malignant solid tumors in children, accounting for 5–6% of all pediatric malignancies, with Wilms’ tumor being the most common renal malignancy [1]. The treatment of Wilms’ tumor is one of the great success stories of oncology. With advances in the diagnosis and multidisciplinary treatment of pediatric malignancies, the prognosis of renal malignancies, especially Wilms’ tumor, has dramatically improved. However, some histologic types have a significantly poorer prognosis, and addressing the predisposing syndrome of Wilms’ tumor and some subgroups with renal malignancies who show a poor prognosis is an important challenge. We have long organized a multidisciplinary team consisting of oncologists, surgeons, pathologists, and subspecialty radiologists for treatment. We have treated pediatric patients with renal tumors such as Wilms’ tumor, clear cell sarcoma of the kidney (CCSK), and rhabdoid tumor of the kidney (MRTK) according to the protocol of the Japanese Wilms Tumor Study (JWiTS) Group [2] which was made based on the National Wilms’ Tumor Study Group (NWTSG) protocol [3]. The purpose of this study is to evaluate the characteristics of renal tumors in patients admitted to our institution between 1984 and 2022, and to report treatment outcomes in order to assess the adequacy of treatment at our institution and to identify problems associated with renal tumors in a variety of children.

## 2. Materials and Methods

The medical records of pediatric patients with renal malignancies who received treatment at our institution between April 1984 and March 2022 were reviewed. The diagnosis was histopathologically confirmed in all patients. The medical charts were reviewed for collection of data on age at the diagnosis, sex, clinical symptoms, associated anomalies, tumor site, and size, surgical intervention, chemotherapy, radiotherapy, pathology (including histological stage), late complications, and prognostic outcomes. Anaplasia was considered to be a histologically “unfavorable” feature, while tumors without anaplasia were considered to be histologically “favorable.” CCSK, MRTK, and renal cell carcinoma (RCC) are considered distinct tumor types, and these were also discussed in this study. In all patients, the clinical stage was determined according to the NWTSG criteria, and was based solely on the anatomic extent of the tumor, without consideration of genetic, biological, or molecular markers. The histologic classification was as defined by the NWTSG study [4].

Event-free survival was defined as the time from study entry to the first occurrence of progression, relapse, and death from any cause, or loss to follow-up. Survival was defined as the time from study entry to death from any cause. Patients without events were censored at the time of their last follow-up examination. The Kaplan–Meier method was used to assess the survival rates.

This study was performed according to the Ethical Guidelines for Medical and Health Research Involving Human Subjects by the Ministry of Health, Labour, and Welfare of Japan in 2014 and complies with the Helsinki Declaration of 1964 (revised in 2013), and was approved by the local ethical committee of our institution (27-119).

## 3. Results

### 3.1. Patients’ Characteristics and Preoperative Data

Twenty-eight pediatric patients with renal malignancies were treated at our institution in this period. A summary of the clinical data of the 28 patients is listed in Table 1.

There were 10 females (35.7%) and 18 males (64.3%). The median age of the patients at the time of diagnosis was 2 years (range, 9 months to 10 years). The site of onset was the right kidney in 11 patients (39.0%), the left kidney in 15 patients (53.5%), and bilateral in 2 patients (7.1%). An abdominal mass or abdominal distension was the most common presenting symptom, occurring in 17 (60.7%) of 28 cases. The other symptoms at the diagnosis were abdominal pain in four patients (14.3%), gross hematuria in four patients (14.3%), varicocele in one patient (3.6%), and the diagnosis was made during the close examination of associated anomalies in three patients (10.7%). Complications included WAGR (Wilms’ tumor, aniridia, genitourinary anomalies, intellectual disability) syndrome in three patients (10.7%), premature chromatid separation/mosaic anomaly syndrome (premature chromatid separation/mosaic variegated aneuploidy syndrome (PCS/MVA syndrome) in one patient (3.6%), horseshoe kidney in one patient (3.6%), and aberrant pancreas in one patient (3.6%).

The pathological diagnosis and stage (National Wilms Tumor Study (NWTS) classification) are shown in Table 2. Twenty-four patients (85.7%) had Wilms’ tumor, and all had favorable histology (FH). Wilms’ tumors were Stage I in six patients (6/24; 25.0%), Stage II in nine patients (9/24; 37.5%), Stage III in five patients (5/24; 20.8%), Stage IV in two patients (2/24: 8.3%), and Stage V in two patients (2/24; 8.3%). Two patients (7.1%) with CCSK both had Stage I disease. One patient had Stage IV MRTK (3.6%), and one patient had Stage IV Mit family translocation RCC (3.6%).

### 3.2. Chemotherapy

Chemotherapy for Wilms’ tumor consisted of vincristine (VCR) and actinomycin D (ACD) for Stage I and Stage II, and VCR, ACD, and doxorubicin (DXR) for Stage III and Stage IV. One Stage III patient with recurrence was treated with irinotecan-temozolomide (IT) after the initial treatment; however, the patient died 3 years after the initial surgery due to pulmonary metastasis. One of the two Stage V patients was a 16-month-old girl with WAGR syndrome who died of massive hemorrhage immediately after starting VCR + ACD. The other patient with PCS/MVA syndrome was started on ACD monotherapy; however, chemotherapy was discontinued due to severe adverse events. Two CCSK patients received VCR + DXR + cyclophosphamide (CPA) + etoposide (VP-16) based on the NWTS5 CCSK regimen. One MRTK patient was treated with VCR + ACD + DXR, but died of respiratory failure due to lung metastasis at 5 months after the start of treatment. One patient with renal cell carcinoma received treatment with a molecular-targeted agent including Sunitinib.

### 3.3. Radiation Therapy

Radiation therapy targeting the primary tumor was administered to 13 (39.3%) patients. Among these, five had Stage III Wilms tumor, two had CCSK, two had Stage II Wilms’ tumor, and four had Stage IV Wilms’ tumor. In one stage IV patient, the tumor had spread into the right atrium at the initial presentation and remained in the inferior vena cava after surgery. One patient with Stage II Wilms’ tumor, who had local recurrence at 4 years after the initial diagnosis, was treated with local radiation therapy.

### 3.4. Late Complications and Prognostic Outcomes

The late complications are shown in Table 2. Chronic renal failure occurred in four patients (14.3%); one of the four patients had bilateral Wilms tumors. Intellectual disability was recognized in three patients (10.7%), two of whom had WAGR syndrome. Small bowel obstruction occurred in two patients (7.1%). Hyperuricemia was present in two patients (7.1%), vagal palsy in one (3.6%), and scoliosis in one (3.6%) who had been treated with radiation therapy. No secondary cancers were observed in any patients.

The survival curves for all renal malignancies are shown in Figure 1. The overall 5-year survival rate was 76.8%. Regarding Wilms’ tumor, survival curves are shown in Figure 2. The 5-year survival rate in all patients with was 85.2%. In patients with stage I–II Wilms’ tumor, the overall 5-year survival rate was 100.0%.

Six deaths occurred during this period, as shown in Table 3: two of the patients who had bilateral Wilms’ tumors with WAGR syndrome, the patient with MRTK, and the Mit translocation RCC died of tumor progression at <1 year after surgery. On the other hand, two deaths occurred without tumor progression. One patient with PCS/MVA syndrome, who had bilateral Wilms’ tumors and underwent bilateral nephrectomy, died of sepsis one year after surgery, while one patient with right unilateral Wilms’ tumor died of strangulated ileus at 11 years after surgery, although she achieved complete remission.

## 4. Discussion

The present retrospective study was conducted at a single center where pediatric malignancy cases in a local area of southern Japan were mostly consolidated. Although the patients’ backgrounds tended to include slightly more boys, the age at onset was similar to previous reports [5]. The most common clinical manifestation was abdominal distention in the literature, which also applied to abdominal mass, which was the most common initial symptom in this study. The frequency of gross hematuria did not differ from that described in the literature [6]. Regarding histology, all Wilms’ tumors showed FH only, which differed from previous reports, while the rates of CCSK, MRTK, and RCC were similar to previous reports [1,7,8].

In principle, surgery and chemotherapy for Wilms’ tumor were based on the NWTS group strategy and regimen at our institution. In this study, one patient with recurrent relapse was treated with IT therapy. Although the first course was unaffected, the patient showed progression during the second course, and died of lung metastasis after the third course [9]. In recent years, there has been one report on the efficacy of IT therapy in combination with VCR and bevacizumab for refractory Wilms’ tumor [5,10]. Various therapies have been tried for refractory Wilms’ tumor, but there is currently no protocol that seems to be sufficiently effective, and treatment is often difficult. Currently, cancer immunotherapy has been tested in clinical trials or basic studies in Wilms’ tumor [11]. New protocols need to be considered to improve survival in these cases [12].

Two patients with CCSK received radiation therapy to the tumor bed with VCR + DXR + CPA + VP-16 based on the NWTS5 regimen [4,13]. Both patients achieved complete remission without any adverse events. In the relevant literature, the 5-year overall survival (OS) and event-free survival (EFS) rates of patients with CCSK were 89% and 79%, respectively, and the 5-year OS and EFS rates reported by the JWiTS group [2], which performed the same treatment as NWTS group were 74.5% and 72.9%, respectively [13]. Both of our cases were Stage I, which may have affected the results, and we have not fully investigated this issue. However, the results of CCSK are slightly lower than those of Wilms’ tumors, so this is an issue for future study [14,15].

In RTK, no effective treatment has been established, and prognosis is very poor: a study of 142 patients with NWTS-1-5 reported that stage and age at diagnosis correlated with prognosis, with an OS rate of 42% for Stage I and II patients, compared to 16% for Stage III and IV [6,16]. The results are not very satisfactory, and in our case, although it was Stage IV, it was refractory and could not save the patient, a fact that seems to go without saying that a new protocol for RTK is needed in the future to improve the survival rate [17].

Regarding irradiation, postoperative radiation therapy targeting the primary tumor is not usually indicated for Stage I or II Wilms’ tumor with FH [4,18]. In the present study, postoperative irradiation of the primary tumor was performed in five patients with Stage III, two patients with CCSK, one patient with Stage II disease with WAGR syndrome, and one patient with a residual tumor in the inferior vena cava. For patients with histological factors associated with a poor prognosis, abdominal irradiation is recommended for CCSK, regardless of stage. MRTK is recommended but the prognosis is extremely poor and the benefit of radiotherapy is unclear [2,8,19].

WAGR syndrome is a syndrome caused by a microdeletion of the short arm 13 region of chromosome 11, which results in Wilms’ tumor following the deletion of the WT1 gene in the same region [20,21]. In the present study, associated anomaly of WAGR syndrome was observed in three patients. One of them had bilateral tumor development. Diseases associated with a high risk of developing bilateral Wilms’ tumor include WAGR syndrome, Denys–Drash syndrome, and Beckwith–Wiedemann syndrome [8,22,23]. These diseases are associated with a high rate of nephrogenic rest (NR), which is considered a precursor lesion to nephroblastoma, and nephroblastomatosis (NBM), in which NR occurs multiply or diffusely [24]. NBM is sensitive to chemotherapy, and some NBM tumors have been reported to shrink or disappear with combination chemotherapy [25]. It has also been reported that chemotherapy can reduce the future incidence of Wilms’ tumor [10]. On the other hand, the risk of developing chronic renal failure as a late complication is high in WAGR syndrome, and one patient was treated with hemodialysis at our institution. According to Breslow et al., 36% of unilateral survivors and 90% of bilateral survivors of Wilms tumor with WAGR syndrome developed end-stage renal failure [12]. In renal biopsy specimens of healthy kidneys in patients with unilateral Wilms’ tumor with WAGR syndrome, the glomeruli were reported to be significantly smaller in comparison to normal glomeruli, which may be a factor in the development of chronic renal failure [26,27]. PCS/MVA syndrome is an autosomal recessive chromosomal instability syndrome associated with growth retardation, microcephaly, and hypercarcinogenicity, and patient cells frequently show premature chromosome segregation (PCS) and multiple aneuploidy mosaicism (MVA). Almost all patients developed Wilms’ tumor or rhabdomyosarcoma [28]. Hanks et al. analyzed the chromosomes of rhabdomyosarcomas that occurred in this disease, and found amplification of chromosomes 3, 8, and 13 and loss of chromosomes 9, 14, and 10, findings that are also seen in sporadic rhabdomyosarcomas [29]. Matsuura et al. speculated that the highly unstable chromosome number in this disease leads to a high rate of development, as well as early development of Wilms’ tumor and rhabdomyosarcoma, because cells with the appropriate chromosome combination for carcinogenesis are likely to develop. The prognosis was considered to be poor, and our case could not be saved.

Renal failure was the most frequent late complication, occurring in 55 of 5823 Wilms’ tumors treated with the NWTS protocol [4]. Renal failure occurred more frequently in bilateral tumors than in unilateral tumors. The most common cause of renal failure was bilateral nephrectomy for treatment-resistant or recurrent tumors. Although the long-term survival rate for patients with Wilms’ tumor has improved, the protocol needs to be changed in order to improve the rate of preservation of the renal parenchyma for patients who are at risk for renal failure. Other late complications include cardiovascular problems as a side effect of DXR, liver damage as a side effect of VCR or ACD, and the development of secondary cancers, none of which were seen in the present study [2].

In Japan, the Japan Wilms Tumor Study Group (JWiTS) was established in 1996, and a nationwide standardized protocol was introduced. As a result, both relapse-free survival (RFS) and overall survival (OS) improved for Wilms tumors at Stages I to III, and the outcomes at our institution were comparable. Currently, our institution has been participating in the International Society of Pediatric Oncology (SIOP) Renal Tumor Group’s collaborative research, including the Umbrella Protocol since 2022, and further advancements are anticipated. However, regarding the treatment strategies for high-risk groups, the current protocols have shown limited efficacy. Despite the presence of a certain proportion of poor-prognosis cases even among Wilms’ tumors, the risk factors for treatment-resistant cases remain unclear. It is imperative to identify novel risk factors that will facilitate the development of new therapeutic approaches in the future.

The present study was associated with some limitations, as it included a relatively small number of patients who were managed at a single institution.

## 5. Conclusions

We reported 28 cases of renal malignancy in our institution. The multidisciplinary teamwork model was feasible, and treatment outcomes were comparable to those in previous reports. However, all stage V patients in our department had FH, but the prognosis was poor due to the complications of PCS/MVA syndrome and WAGR syndrome. In advanced cases, the prognosis of MRTK was still poor, suggesting the need to establish a different therapeutic protocol.

## Figures and Tables

**Figure 1 diseases-13-00040-f001:**
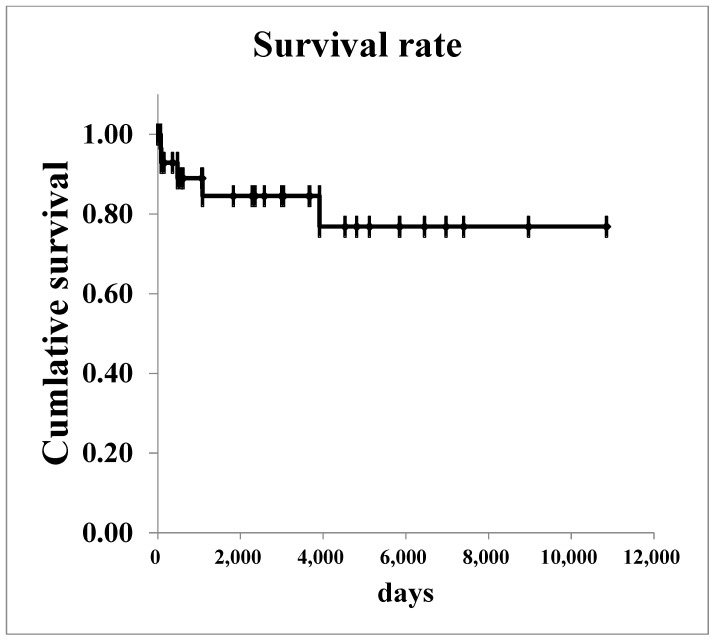
The survival curves of all renal malignancies.

**Figure 2 diseases-13-00040-f002:**
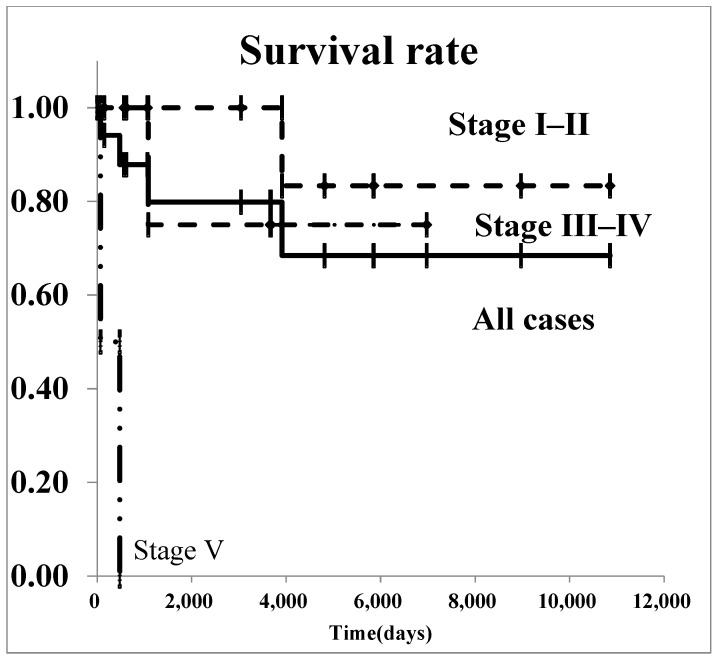
The survival curves of Wilms’ tumor by stage.

**Table 1 diseases-13-00040-t001:** Summary of clinical data in 28 patients with renal malignancies.

			n
Sex (male:female)			18:10 (64.3%:35.7%)
Age at diagnosis			0 y 1 m–10 y 6 m (Median: 2 y)
Laterality	Right		11 (39.3%)
	Left		15 (53.6%)
	Bilateral		2 (7.1%)
Symptoms at diagnosis	Abdominal pain		17 (60.7%)
	Abdominal distension		5 (17.9%)
	Hematuria		6 (21.4%)
	Varicocele		1 (3.6%)
	Incidental		4 (14.3%)
Associated anomalies	WAGR syndrome		3 (10.7%)
	PCS/MVA syndrome		1 (3.6%)
	Horseshoe kidney		1 (3.6%)
	Aberrant pancreas		1 (3.6%)
Pathological diagnosis	Wilms tumor (n = 24)	Stage I	6 (25.0%)
		Stage II	9 (37.5%)
		Stage III	5 (20.8%)
		Stage IV	2 (8.3%)
		Stage V	2 (8.3%)
	CCSK (n = 2)	Stage I	2 (7.1%)
	RTK (n = 1)	Stage IV	1 (3.6%)
	RCC (n = 1)	Stage IV	1 (3.6%)

Abbreviation: CCSK: clear cell sarcoma of the kidney; MRTK: malignant rhabdoid tumor of the kidney; RCC: renal cell carcinoma.

**Table 2 diseases-13-00040-t002:** Summary of late complications.

	n	
Chronic renal failure	4	(14.3%)
Intellectual disability	3	(10.7%)
Small bowel obstruction	2	(7.1%)
Hyperuricemia	2	(7.1%)
Vagal palsy	1	(3.6%)
Scoliosis	1	(3.6%)

**Table 3 diseases-13-00040-t003:** A summary of mortality cases.

Case of No.	Age at Diagnosis	Pathology	Stage	Laterality	Associated Anomalies	Cause of Death
1	2 y 4 m	Wilms tumor, FH	2	Left	None	Strangulated ileus
2	3 y 4 m	Wilms tumor, FH	3	Right	WAGR syndrome	Tumor progression, Lung metastasis
3	1 y 4 m	Wilms tumor, FH	5	Bilateral	WAGR syndrome	Tumor progression
4	7 m	Wilms tumor, FH	5	Bilateral	PCS/MVA syndrome	Sepsis
5	6 m	RTK	4	Right	none	Tumor progression
6	10 y	RCC	4	Left	none	Tumor progression

Abbreviation: FH: favorable histology;RTK: rhabdoid tumor of the kidney; RCC: renal cell carcinoma.

## Data Availability

The data presented in this study are available on request from the corresponding author.

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
