# Peer review of "A Retrospective Review of 28 Cases of Pediatric Malignant Renal Tumors at a Single Institution"

_diseases, 2025, doi:10.3390/diseases13020040_

Round 1
Reviewer 1 Report
Comments and Suggestions for Authors
Minor grammar and editorial suggestions throughout. See attached version for comments and suggestions.
The paper was well laid out and described the cases well. However, after reading, it feels like there is still some context missing. It would be helpful to add more context to why these cases were reported and how the report of these cases fit into the bigger picture. While it's good that what is reported in these 28 cases seems to be in line with what has already been reported, is there anything new here or is there something here that might help other researchers put their own findings into perspective? Was there any change in treatment approaches over time? A little more context for putting this in the bigger picture would be good.

Author Response
"Minor grammar and editorial suggestions throughout. See attached version for comments and suggestions.
The paper was well laid out and described the cases well. However, after reading, it feels like there is still some context missing. It would be helpful to add more context to why these cases were reported and how the report of these cases fit into the bigger picture. While it's good that what is reported in these 28 cases seems to be in line with what has already been reported, is there anything new here or is there something here that might help other researchers put their own findings into perspective? Was there any change in treatment approaches over time? A little more context for putting this in the bigger picture would be good."
 Reply: Thank you very much for your comments and suggestions on our manuscript. In accordance with the comments, we revised manuscript.
Reviewer 2 Report
Comments and Suggestions for Authors
Dear Authors,
This work presents interesting data on the characteristics of renal tumors. However, several points require clarification and enhancement to improve its impact.
-
Terminology: In the abstract, "MRT" is used in line 33, while "MTRK" appears in line 49. Please ensure consistent use of terminology throughout the manuscript.
-
Literature Review: The literature review should be updated to include recent findings (up to 2024) to provide a more comprehensive and current perspective.
-
Study Purpose and Comparison: The stated purpose of the study (lines 51-52) is unclear.
- If the aim was to describe the morphological characteristics of renal tumors, a comparison with data from other centers in your country or internationally over the same period would significantly enhance the study's impact.
- If the study aimed to analyze treatment outcomes, the manuscript should explicitly address how treatment approaches have evolved over the nearly 40-year study period and how these changes have influenced treatment outcomes.
-
Impact of Treatment Evolution: The significant changes in treatment protocols since 1988 likely influenced the observed outcomes, such as complications (Table 3) and causes of death (Table 4). Please discuss how these evolving treatment paradigms have impacted the observed trends.
-
Treatment Effectiveness by Tumor Type: An analysis of treatment effectiveness based on the specific morphological type of renal tumor would be valuable and would further enhance the clinical significance of the study.
-
Clinical Implications: Please include a dedicated section discussing the practical implications of your findings for clinicians. What are the key takeaways for clinicians based on your study results? How can these findings inform their clinical practice and patient management?
We encourage you to address these points and revise the manuscript accordingly.
Author Response
- “Terminology: In the abstract, "MRT" is used in line 33, while "MTRK" appears in line 49. Please ensure consistent use of terminology throughout the manuscript.”
Reply: Thank you very much for your comments and suggestions on our manuscript. In accordance with the comments, all “MRT” was changed into “MRTK”
- Literature Review: The literature review should be updated to include recent findings (up to 2024) to provide a more comprehensive and current perspective.
Currently, cancer immunotherapy has been tested in clinical trials or basic studies in WT.
- Study Purpose and Comparison: The stated purpose of the study (lines 51-52) is unclear.
Reply: Thank you very much for your comments.
Following change was made.
The purpose of this study is to evaluate the characteristics of renal tumors in patients admitted to our institution between 1984 and 2022 and to report treatment outcomes in order to assess the adequacy of treatment at our institution and to identify problems associated with renal tumors in a variety of children.
- If the aim was to describe the morphological characteristics of renal tumors, a comparison with data from other centers in your country or internationally over the same period would significantly enhance the study's impact.
Reply: Thank you very much for your comments.
In “Discussion” section, I have added a reference from the Japanese pediatric renal tumor group.
And
Regarding histology, all Wilms' tumors showed FH, regardless of stage, and the rates of CCSK, MRTK, and RCC were similar to previous reports
Was changed to
Regarding histology, all Wilms' tumors showed FH only, which differed from previous reports, while the rates of CCSK, MRTK, and RCC were similar to previous reports
- If the study aimed to analyze treatment outcomes, the manuscript should explicitly address how treatment approaches have evolved over the nearly 40-year study period and how these changes have influenced treatment outcomes.
- Impact of Treatment Evolution: The significant changes in treatment protocols since 1988 likely influenced the observed outcomes, such as complications (Table 3) and causes of death (Table 4). Please discuss how these evolving treatment paradigms have impacted the observed trends.
Reply: Thank you very much for your comments.
However, the treatment strategy follows that of the Pediatric Renal Tumor Group in Japan. The initial chemotherapy itself remains the almost same, especially for Wilms' tumor, because of its good outcome.
- Treatment Effectiveness by Tumor Type: An analysis of treatment effectiveness based on the specific morphological type of renal tumor would be valuable and would further enhance the clinical significance of the study.
Reply: Thank you very much for your comments.
- Clinical Implications: Please include a dedicated section discussing the practical implications of your findings for clinicians. What are the key takeaways for clinicians based on your study results? How can these findings inform their clinical practice and patient management?
In Japan, the Japan Wilms Tumor Study Group (JWiTS) was established in 1996, and a nationwide standardized protocol was introduced. As a result, both relapse-free survival (RFS) and overall survival (OS) improved for Wilms tumors at Stages I to III, and the out-comes at our institution were comparable. Currently, our institution has been participat-ing in the International Society of Pediatric Oncology (SIOP) Renal Tumor Group's collab-orative research, including the Umbrella Protocol since 2022, and further advancements are anticipated. However, regarding the treatment strategies for high-risk groups, the cur-rent protocols have shown limited efficacy. Despite the presence of a certain proportion of poor-prognosis cases even among Wilms tumors, the risk factors for treatment-resistant cases remain unclear. It is imperative to identify novel risk factors that will facilitate the development of new therapeutic approaches in the future.
Round 2
Reviewer 1 Report
Comments and Suggestions for Authors
All original comments adequately addressed. The addition of the context statement in the introduction and the paragraph in the discussion were both helpful in providing context for the study and report as a whole.
Reviewer 2 Report
Comments and Suggestions for Authors
Thank you for your replies